# Prevalence and outcomes of patients developing heparin-induced thrombocytopenia during extracorporeal membrane oxygenation

**Matthias Lubnow** [1][*], **Johannes Berger**[1], **Roland Schneckenpointner**[1], **Florian Zeman**[2], **Dirk Lunz**[3], **Alois Philipp**[4], **Maik Foltan**[4], **Karla Lehle**[4], **Susanne Heimerl**[5], **Christina Hart**[6], **Christof Schmid**[4], **Christoph Fisser**[1][‡], **Thomas Müller**[1][‡]

1 Department of Internal Medicine II, University Hospital Regensburg, Regensburg, Bavaria, Germany,
2 Center for Clinical Studies, University Medical Center Regensburg, Regensburg, Bavaria, Germany,
3 Department of Anesthesiology, University Hospital Regensburg, Regensburg, Bavaria, Germany,
4 Department of Cardiothoracic Surgery, University Hospital Regensburg, Regensburg, Bavaria, Germany,
5 Department of Clinical Chemistry and Laboratory Medicine, University Hospital Regensburg, Regensburg, Bavaria, Germany, 6 Department of Internal Medicine III, University Hospital Regensburg, Regensburg, Bavaria, Germany

☯ These authors contributed equally to this work.
‡ CF and TM also contributed equally to this work.
\* Matthias.Lubnow@ukr.de

**Data Availability Statement:** There are ethical and legal restrictions by german law and the local ethics committee on sharing a de-identified data

## Abstract

### Objectives

Unfractionated heparin (UFH) is the commonly used anticoagulant to prevent clotting of the ECMO circuit and thrombosis of the cannulated vessels. A side effect of UFH is heparin-induced thrombocytopenia (HIT). Little is known about HIT during ECMO and the impact of changing anticoagulation in ECMO patients with newly diagnosed HIT. The aim of the study was to determine the prevalence, complications, impact of switching anticoagulation to argatroban and outcomes of patients developing heparin-induced thrombocytopenia (HIT) during either veno-venous (VV) or veno-arterial (VA) ECMO.

### Methods

Retrospective observational single centre study of prospectively collected data of consecutive patients receiving VV ECMO therapy for severe respiratory failure and VA ECMO for circulatory failure from January 2006 to December 2016 of the Medical intensive care unit (ICU) of the University Hospital of Regensburg. Treatment of HIT on ECMO was done with argatroban.

### Results

507 patients requiring ECMO were included. Further HIT-diagnostic was conducted if HIT-4T-score was ≥4. The HIT-confirmed group had positive HIT-enzyme-linked-immunosorbent-assay (ELISA) and positive heparin-induced-platelet-activation (HIPA) test, the HIT-

set publicly as it potentially contains identifying or sensitive information. Data access on reasonable request can be provided by the corresponding author Matthias Lubnow, Department of Internal Medicine II, University Medical Center Regensburg, Franz-Josef-Strauss-Allee 11, D-93053 Regensburg, Germany. Email: Matthias. Lubnow@ukr.de or the Ethics Committee of the University Regensburg: Universität Regensburg Ethikkommission, D-93040 Regensburg, Germany. Email: ethikkommission@ur.de.

**Funding:** The authors received no specific funding for this work.

**Competing interests:** I have read the journal's policy and the authors of this manuscript have the following competing interests: ML received lecture honoraria from Fresenius Medical Care. This does not alter our adherence to PLOS ONE policies on sharing data and materials.

suspicion group a positive HIT-ELISA and missing HIPA but remained on alternative anticoagulation until discharge and the HIT-excluded group a negative or positive HIT-ELISA, however negative HIPA. These were compared to group ECMO-control without any HIT suspicion. The prevalence of HIT-confirmed was 3.2%, of HIT-suspicion 2.0% and HIT-excluded 10.8%. Confirmed HIT was trendwise more frequent in VV than in VA (3.9 vs. 1.7% p = 0.173). Compared to the ECMO control group, patients with confirmed HIT were longer on ECMO (median 13 vs. 8 days, p = 0.002). Different types of complications were higher in the HIT-confirmed than in the ECMO-control group, but in-hospital mortality was not different (31% vs. 41%, p = 0.804).

## Conclusion

HIT is rare on ECMO, should be suspected, if platelets are decreasing, but seems not to increase mortality if treated promptly.

## Introduction

The use of extracorporeal membrane oxygenation (ECMO) is increasing steadily in recent years [1]. Unfractionated heparin (UFH) is the commonly used anticoagulant to prevent clotting of the ECMO circuit and thrombosis of the cannulated vessels [2]. A well-known side effect of UFH is the development of heparin-induced thrombocytopenia (HIT) leading to thromboembolism, thrombocytopenia, and bleeding. HIT has been shown to result in mortality rates between 10 to 30% [3, 4].

ECMO therapy is mainly used as a salvage therapy for patients with life-threatening disease [5] and can itself result in venous or arterial thrombotic events, clotting of the oxygenator combined with bleeding events [6–8] by activation and consumption of platelets, thereby mimicking HIT [9, 10]. Irrespective of HIT, these disorders are linked to an increased mortality [11]. Additionally, ECMO and other circulatory assist devices may promote HIT development as a result of persistent platelet activation and platelet-factor-4 release [12–14], but the impact of changing anticoagulation in ECMO patients newly diagnosed with HIT is still unclear.

Although the prevalence of HIT in ECMO patients might be increased compared to other critical ill patients due to the inherent risks of extracorporeal circulation, data on this important subject are limited to small studies, case reports and mainly focused on VA ECMO [15–18]. A recent multicentre study evaluating HIT in patients requiring ECMO support included only patients with VA ECMO [20].

Therefore, we undertook this study to determine the prevalence, risk factors, complications, the impact of switching anticoagulation to argatroban and outcomes of patients developing HIT during either veno-venous (VV) or veno-arterial (VA) ECMO therapy.

## Materials and methods

### Study subjects

This analysis is a retrospective observational single centre study of prospectively collected data (Regensburg ECMO Registry) at the medical intensive care unit (ICU) of the University Hospital of Regensburg. All consecutive patients admitted from January 2006 to December 2016 for severe respiratory failure (PaO$_2$/FiO$_2$ < 85mmHg and/or refractory respiratory acidosis with pH < 7.25) receiving VV ECMO therapy and patients receiving VA ECMO for circulatory failure including those undergoing extracorporeal cardio-pulmonary resuscitation

(ECPR) were eligible for this analysis. The ECMO systems used are listed in the S1 File. Since early 2017, argatroban has been increasingly used as an anticoagulant in VV ECMO therapy, so no further patient recruitment has occurred. Routine data such as demographics, clinical characteristics, laboratory and respiratory parameters, ECMO specifics and cerebral performance category (CPC) after decannulation [19] were extracted from the hospital's electronic patient data management system and the Regensburg ECMO registry. Due to the development of HIT antibodies within 5 to 10 days after exposure to UFH [3, 14], laboratory parameters were compared from 5 days before to 7 days after the first suspicion of HIT [20].

The requirement of individual patient consent and necessity of approval for the data report was waived by the ethics committee of the University of Regensburg (ethics statement No.:18-1062-104) because of the study's design and data collection from routine care.

## Anticoagulation strategy

All ECMO patients received as a standard of care UFH with a goal activated partial thromboplastin time (aPTT) of 50 sec in VV ECMO and of 60 sec in VA ECMO. In case of HIT-suspicion anticoagulation was changed to argatroban with aPTT goal of 50 sec, in case of confirmed HIT with an aPTT-goal of 60 sec (S3 Fig). In case of severe thrombocytopenia or bleeding lower individual aPTT-goals were accepted.

## Screening algorithm for heparin-induced thrombocytopenia

All patients anticoagulated with UFH and a sudden decrease in platelets of more than 30% after 5 days on ECMO without other explaining pathology and/or thrombotic complications were screened for HIT according to the HIT-4T-Score [21]. The HIT-4T-score includes extent of thrombocytopenia, timing of onset, thrombosis and other possible causes of thrombocytopenia. Details of the HIT-4T-score are available in the S1 File and have been previously published [22]. Other reasons for a drop in platelet count like sepsis, disseminated intravascular coagulation (DIC), drug reactions or pulmonary embolism [3] were excluded by review of clinical / laboratory parameters; appropriate imaging was carried out if necessary.

Diagnosis of HIT was made by detection of antibodies against platelet factor 4 using enzyme-linked immunosorbent assay (ELISA) and by using a positive heparin-induced platelet activation (HIPA) test for confirmation (details see S1 File). According to the results of the HIT diagnostics, patients were divided into four different groups: "HIT-confirmed" (HIT-4T-Score ≥ 4, positive HIT ELISA and positive HIPA test), "HIT-suspicion" (HIT-4T-Score ≥ 4, positive HIT-ELISA, missing HIPA test) and "HIT-excluded" (HIT-4T-Score ≥ 4, negative or positive HIT-ELISA but negative HIPA test).

The reason for the absence of the HIPA test in the HIT-suspicion group was usually that it was forgotten to be sent or that it was temporarily unavailable, therefore no specific preselection of this patient group has occurred.

All patients with clinical suspicion of HIT (HIT-4T-Score ≥ 4) were switched from UFH to argatroban. Anticoagulation with argatroban was continued in the HIT-confirmed and the HIT-suspicion group and switched back to heparin in the HIT-excluded group. Lastly, the group ECMO-control was defined as those patients receiving ECMO therapy without clinical suspicion of HIT (HIT-4T-Score < 4) and without change of anticoagulation.

## Complications during ECMO therapy

Complications were identified using the electronic patient data management system and were divided into bleeding, thromboembolic and technical complications. All complications were verified by clinical and laboratory parameters, ultrasound and computer tomography [6].

Bleeding complications were subclassified according to the location of bleeding (cerebral, pulmonary, gastrointestinal, wound, diffuse, retroperitoneal, cannula insertion sites, oral and others) [21]. Thrombotic events were classified into arterial thrombotic events and sub-classified as intracardiac thrombosis, cerebral ischemia, ischemia of cannulated leg or other arteries. Similarly, venous thrombotic events were assessed by the location and classified according to the cannulated vessel, vena cava inferior, pulmonary embolism and other veins. Routine ultrasound as a standard to exclude venous thrombosis during and after the ECMO therapy was established in November 2014. Thus, the control group according to venous thrombosis consisted of 251 patients (VV: 171 vs. VA: 80), whereas patients with suspected HIT were investigated thoroughly for complications during the entire duration of the study. Technical problems were classified as previously published into pump head thrombosis, oxygenator or circuit thrombosis, cannula thrombosis, plasma leak of the oxygenator and impaired gas transfer [23].

## Statistical methods

All quantitative data are expressed as median (interquartile range) or absolute and relative frequencies. Differences between groups were assessed using the Mann-Whitney-U-Test or Kruskal-Wallis-Test followed by Dunn's test for multiple pairwise comparisons in case of significance or by using a chi-squared test of independence for nominal variables. All reported p-values were two-sided, and a p-value of $\leq 0.05$ was considered statistically significant. All analyses were performed using R (version 4.0.1) (R Foundation for Statistical Computing, Vienna, Austria) and IBM SPSS Statistic software version 25.0 (SPSS Inc. Chicago, IL, USA).

## Results

### Study population

The study population included 507 patients, 176 patients with VA ECMO (35%) and 331 with VV ECMO (65%). 426 patients (84%, 260 VV ECMO and 166 VA ECMO) showed no significant decrease in platelet count after more than 5 days on ECMO and therefore not triggering further HIT diagnostics (group ECMO-control). According to the HIT-4T-Score, diagnosis of HIT was suspected in 81 patients (16%). Of those, HIT was confirmed in 16 cases (20%) with significant drop of thrombocytes >5 days after start of ECMO therapy, a positive HIT ELISA and HIPA test (group HIT-confirmed) and excluded in 55 (68%). Ten patients (12%) had a positive HIT ELISA and an alternative anticoagulation until discharge, but missing HIPA confirmation test (group HIT-suspicion).

### Prevalence of HIT and time of onset

With respect to the complete study population, the prevalence of confirmed HIT was 3.2% (Fig 1). The prevalence of confirmed HIT was trendwise higher in VV ECMO than in VA ECMO (3.9% [13/331] vs. 1.7% [3/176]), p = 0.173. Clinical characteristics of these 16 patients are presented in S1 Table. Similarly, in the group HIT-suspicion were trendwise more patients supported with VV than with VA ECMO (2.7% [9/331] vs. 0.6% [1/176]), p = 0.097. Combining both groups, the prevalence of HIT suspicion and confirmed HIT on VV ECMO is 6.6% (22/331) and 2.3% (4/176) on VA ECMO (p = 0.034). Severity of critical illness was not different between groups (Table 1). The prevalence in relation to ECMO therapy duration of HIT-confirmed was 0.48/10days on VV ECMO and 0.39/10days of VA ECMO (p = 0.439) whereas of HIT-suspicion 0.32/10days on VV ECMO and 0.056/10days on VA ECMO (p = 0.200). No difference was seen in the temporal occurrence of HIT between VV and VA ECMO (p = 0,145) (Fig 2). Median time between start of ECMO therapy and HIT was 7,5 days in

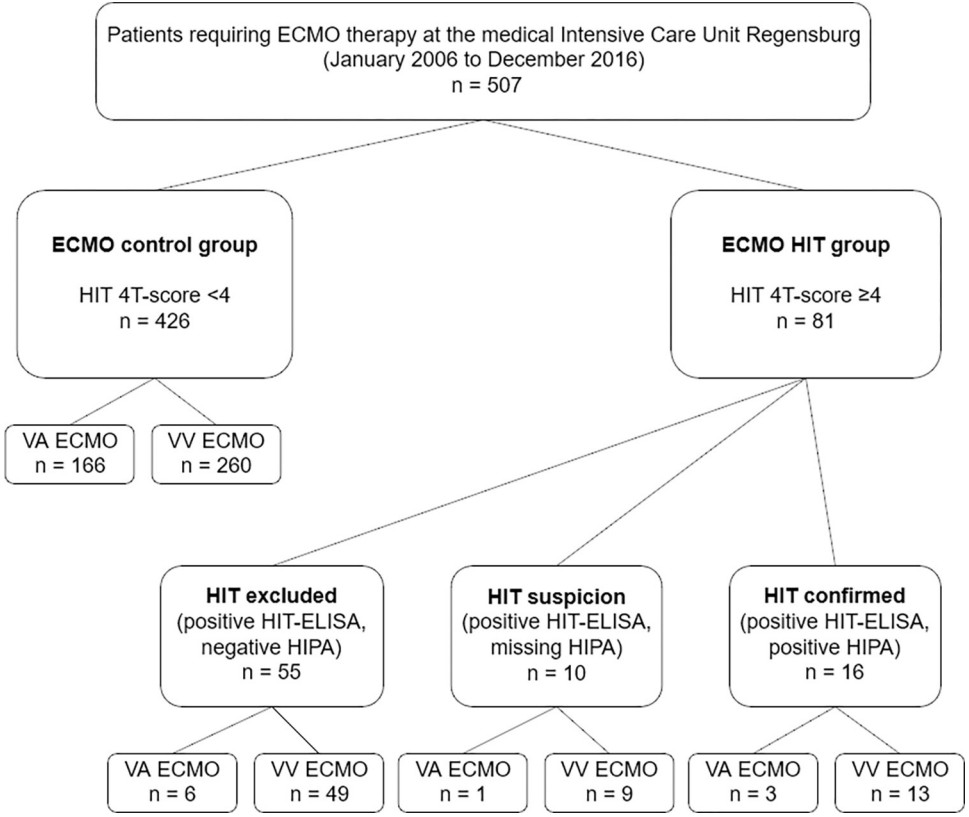

**Fig 1. Flowchart of the study.** Flowchart of the observational study evaluating heparin-induced thrombocytopenia (HIT) of the prospective extracorporeal membrane oxygenation (ECMO) registry Regensburg. VA: veno-arterial; VV: veno-venous; ELISA: enzyme-linked immunosorbent assay; HIPA: heparin induced platelet aggregation.

confirmed HIT. Three patients developed confirmed HIT in less than 5 days after ECMO start, none of the patients after the 12th day (S1 Table).

## Laboratory parameters

At admission, platelets were not different between groups, but the nadir was significantly lower in those groups with initial suspicion of HIT compared to the ECMO control group, which results also from the study design as groups were initially defined based on the 4T-score. Trajectories of mean thrombocyte counts of the different groups are presented in Fig 3 and baseline characteristics in Table 1. Trajectories of additional laboratory parameters such as D-Dimers, fibrinogen, plasma free hemoglobin, leucocytes, aPTT, CRP and antithrombin showed no differences between groups but the group HIT-suspicion had higher fibrinogen and CRP levels compared to the other groups (S1–S7 Figs).

## Complications

Complications of the different groups during the ECMO therapy are shown in Table 2. The group HIT-confirmed had numerically a higher thrombosis rate than the group ECMO-control (details for arterial and venous thrombotic events see S2 Table). Similarly, technical problems were more frequent in the group HIT-confirmed and are given in more detail in S3 Table. A higher prevalence of bleeding was observed in those groups that were at least temporarily treated with argatroban (Tables 2 and S4). Also, the consumption of blood products was

**Table 1. Patient Characteristics at initiation of extracorporeal membrane oxygenation and outcome.**

| Patient characteristics | HIT-confirmed n = 16 | HIT-suspicion n = 10 | HIT-excluded n = 55 | ECMO-control n = 426 | Global P-value |
|---|---|---|---|---|---|
| Age (years) | 55.2 (46–65) | 55.5 (41–65) | 52.6 (40–63) | 54.8 (45–64) | 0.885 |
| Gender (male) n (%) | 12 (75) | 4 (40) | 32 (58) | 285 (67) | 0.158 |
| Body mass index, kg/m$^2$ | 27.8 (24–34) | 33.9 (26–38) | 28.0 (25–34) | 27.7 (24–31) | 0.233 |
| SOFA | 12.0 (9–14) | 11.5 (10–13) | 11.0 (8–15) | 12.0 (9–15) | 0.818 |
| VA ECMO n (%) | 3 (19) | 1 (10) | 6 (11) | 166 (39) | <0.001[3] |
| Time between intubation and ECMO, days | 1.0 (1–2) | 4.0 (1–9) | 1.5(0–9) | 1.0 (0–2) | <0.002 [2,3] |
| Days on ECMO | 12.5 (9–26) | 16.0 (12–21) | 14.0 (10–24) | 8.0 (4–14) | <0.001[1,2,3] |
| Days from ECMO start to HIT suspicion / change to alternative anticoagulation | 7.5 (5–11) | 12.0 (9–15) | 6.0 (2–12) | - | 0.014[4,5] |
| Thrombocytes start ECMO, /nL | 241 (135–334) | 185 (64–320) | 218 (152–303) | 187 (117–268) | 0.406 |
| Thrombocytes nadir, /nL | 46 (32–81) | 36 (21–60) | 47 (22–74) | 113 (53–175) | <0.001[3] |
| MAP | 67.0 (54–75) | 64.0 (60–66) | 68.0 (62–79) | 65.0 (53.72) | 0.023[3] |
| pH | 7.24(7.1–7.3) | 7.22 (7.1–7.3) | 7.25 (7.2–7.4) | 7.19 (7.1–7.3) | 0.033[3] |
| pCO$_2$, mmHg | 58.5 (43–92) | 64.5 (53–77) | 65.0 (51–79) | 63.0 (50–80) | 0.978 |
| PaO$_2$/FiO$_2$, mmHg | 77.0 (63–125) | 76.5 (66–108) | 65.0 (57–89) | 68.0 (55–96) | 0.280 |
| Minute ventilation, L/min | 12.5 (10–17) | 12.2 (7–13) | 10.5 (9–14) | 10.0 (7–12) | <0.001[1,3] |
| Positive inspiratory pressure, cmH$_2$O | 33.0 (26–39) | 37.5 (32–42) | 35.0 (31–38) | 32.0 (26–36) | 0.007[2,3] |
| Positive end-expiratory pressure, cmH$_2$O | 15.0 (11–16) | 20.0 (11–24) | 15.0 (13–20) | 14.0 (10–16) | 0.019[3] |
| **Clinical outcome** | | | | | |
| ICU length of stay, d | 34 (19–38) | 43 (26–46) | 30 (21–43) | 23 (15–36) | 0.983 |
| Hospital length of stay, d | 35 (20–45) | 41.5 (18–55) | 36 (28–54) | 32 (21–48) | 0.934 |
| Inhospital mortality n (%) | 5 (31) | 3 (30) | 22 (40) | 173 (41) | 0.804 |
| ICU mortality n (%) | 5 (31) | 3 (30) | 16 (29) | 170 (40) | 0.977 |
| Mortality on ECMO n (%) | 4 (25) | 3 (30) | 16 (29) | 128 (30) | 0.383 |
| CPC scale | 1 (1–1.3) | 1 (1–1.3) | 1 (1.0) | 1 (1.0) | 0.923 |

Data are expressed as n (%) or median (interquartile range:q1-q3); HIT: heparin-induced thrombocytopenia, ECMO: extracorporeal membrane oxygenation; SOFA: sequential organ failure assessment; MAP: mean arterial pressure; ICU: intensive care unit; CPC scale: cerebral performance category scale; Post-hoc pairwise comparisons with significance (p<0.05)

[1] HIT-confirmed vs. ECMO-control

[2] HIT-suspicion vs. ECMO-control

[3] HIT-excluded vs. ECMO-control

[4] HIT-confirmed vs. HIT-suspicion

[5] HIT-suspicion vs. HIT-excluded.

different between groups with lowest requirement of packed red blood cells in the ECMO-control group (S5 Table).

All groups showed a decrease in platelet count over the first 5 days on ECMO (Fig 3). As easy accessible marker to raise suspicion of HIT on ECMO a further decrease of platelets of more than 30% after the fifth day of ECMO therapy was evaluated. This further decrease differentiated between group ECMO-control and the groups with initial suspicion of HIT (p < 0.001) but there was no difference between the combined group of HIT-confirmed and HIT-suspicion vs. the HIT-excluded group (p = 0.053).

The nadir of the thrombocytes in the group HIT-confirmed, the group HIT-suspicion and the group HIT-excluded was statistically different to the group ECMO-control after the 7$^{th}$ day on ECMO (46x10$^9$/L vs. 36x10$^9$/L vs. 47x10$^9$/L vs. 113x10$^9$/L, p<0.001, respectively). The decline of thrombocytes was more pronounced in VA compared to VV ECMO (S8 Fig).

## Time to HIT in VV and VA ECMO patients of group HIT-confirmed

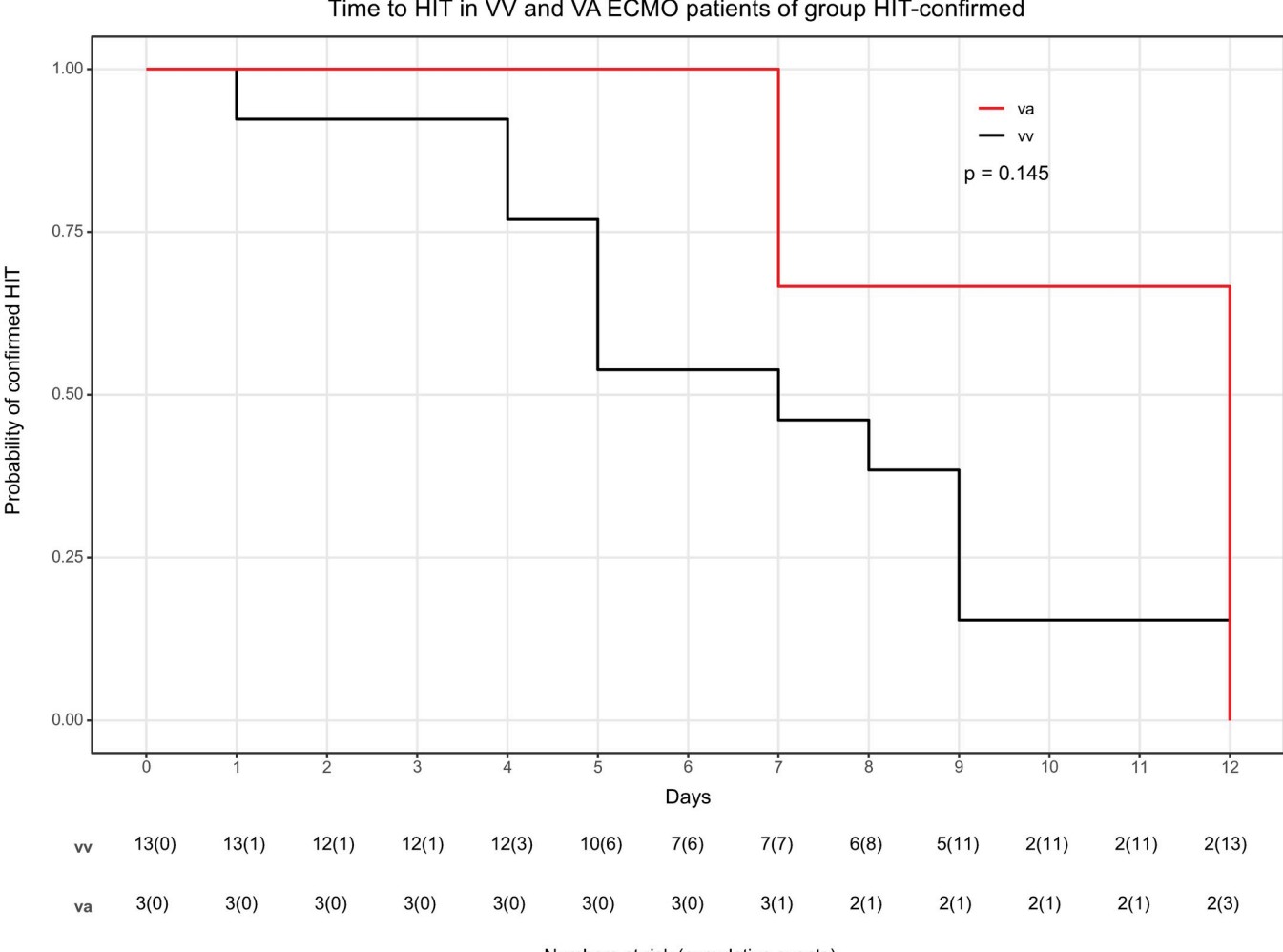

| | | | | | | | | | | | | |
|---|---|---|---|---|---|---|---|---|---|---|---|---|
| vv | 13(0) | 13(1) | 12(1) | 12(1) | 12(3) | 10(6) | 7(6) | 7(7) | 6(8) | 5(11) | 2(11) | 2(11) | 2(13) |
| va | 3(0) | 3(0) | 3(0) | 3(0) | 3(0) | 3(0) | 3(0) | 3(1) | 2(1) | 2(1) | 2(1) | 2(1) | 2(3) |

Numbers at risk (cumulative events)

**Fig 2. Confirmed HIT probability during ECMO therapy according to VV and VA ECMO.** Kaplan-Meier plot of HIT probability during ECMO therapy according to VV ECMO and VA ECMO of group HIT-comfirmed. X-axis: Time to HIT = Time between ECMO initiation and switch to alternative anticoagulation (argatroban). Y-axis: Probability of confirmed HIT diagnosis 0–100%.

After changing to argatroban in all groups but the group ECMO-control, the number of thrombocytes increased (Fig 3). The thrombocyte count seven days after change of anticoagulation was 104 x $10^9$/L, 162 x $10^9$/L and 103 x $10^9$/L in the HIT-confirmed, the HIT-suspicion group and the HIT-excluded group.

If after a HIT diagnosis, while being on argatroban, an oxygenator exchange was necessary, the use of a heparin coated system did not cause another decrease in platelet count nor did it result in higher complication rates (S9 and S10 Figs). Noteworthy, two patients on a non-heparin-coated system did not show a faster increase of thrombocytes after change to argatroban compared to heparin-coated systems (S11 Fig).

## Outcome

Patients developing HIT during ECMO support spent a longer time on ECMO compared to the control group (13 days (IQR: 9–26) vs. 8 (IQR 4–14); p<0.001), but no differences in survival (in hospital mortality: group HIT-confirmed 31% vs. 41% ECMO-control, p = 0.804), length of stay, or CPC scale were observed between groups (Table 1).

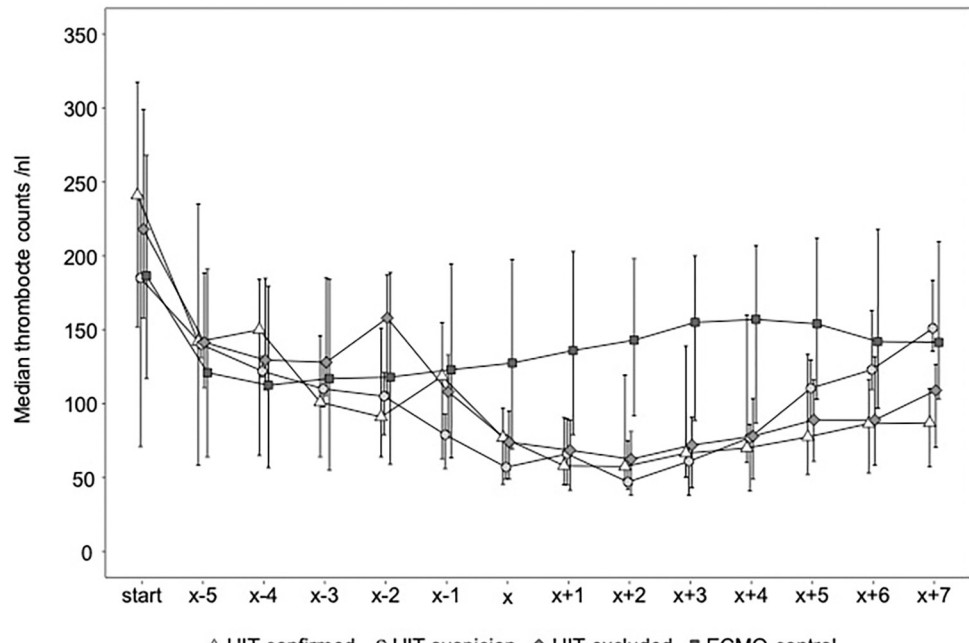

**Fig 3. Platelet count on extracorporeal membrane oxygenation.** Trajectories of thrombocytes before and after suspicion of heparin-induced thrombocytopenia (HIT) according to the groups: HIT-confirmed, HIT-suspicion, HIT-excluded and ECMO-control. Data show median and interquartile range (q1-q3). Time axis in days from day x. x: day of HIT suspicion (change to alternative anticoagulation) or, for group ECMO-control day 7 of ECMO therapy (as median time to HIT on ECMO was 7,5 days). 35 patients were excluded because the extracorporeal membrane oxygenation (ECMO) was explanted within 3 days after changing of anticoagulation or they died within 3 days after changing of anticoagulation, to show the effect of the alternative anticoagulation on coagulation parameters. Occasional missings e.g. if ECMO duration was shorter than 7 days.

**Table 2. Complications on extracorporeal membrane oxygenation.**

|  | HIT-confirmed n = 16 (VV/VA: 13/3) | HIT-suspicion n = 10 (VV/VA: 9/1) | HIT-excluded n = 55 (VV/VA: 49/6) | ECMO-control n = 251 (VV/VA: 171/80) | Global P-value |
|---|---|---|---|---|---|
| Bleeding, n (%) | 6 (38) | 3 (30) | 22 (38) | 54 (22) | 0.054 |
| Several Bleedings, n (%) | 2 (13) | 3 (30) | 4 (7) | 9 (4) | 0.002[1,3] |
| Technical problem, n (%) | 6 (38) | 1 (10) | 24 (44) | 51 (20) | <0.001[1,2] |
| Several technical problems, n (%) | 1 (6) | 0 (0) | 10 (20) | 30 (12) | 0.136 |
| Thrombotic event, n (%) | 10 (63) | 4 (40) | 26 (47) | 143 (56) | 0.330 |
| Several thrombotic events, n (%) | 4 (25) | 3 (30) | 9 (18) | 30 (12) | 0.307 |
| AKI on ECMO, n (%) | 3 (19) | 3 (30) | 11 (20) | 94 (22) | 0.897 |

Data are expressed as n (%) or median (interquartile range: q1-q3); HIT: heparin-induced thrombocytopenia, VV: veno-venous; VA: veno-arterial; ECMO: extracorporeal membrane oxygenation; several bleedings/technical problems/thrombotic events: number of patients with more than one respective event during ECMO therapy; AKI: acute kidney injury, defined as need for dialysis/CRRT. Post-hoc pairwise comparisons with significance (p<0.05)

[1] HIT suspicion vs. ECMO control

[2] excluded HIT vs. ECMO control

[3] HIT suspicion vs. excluded HIT. 175 patients from group ECMO control were excluded in this analysis, because standardized screening for thrombotic complications without any HIT suspicion was only established in 2014.

## Discussion

This is the largest single centre study in a medical ICU ECMO cohort differentiated in VV and VA ECMO providing novel insights into prevalence, risk factors, complications and outcome of patients developing HIT while on ECMO support. First, the prevalence of HIT was 3.2% irrespective of ECMO mode, 3.9% in VV and 1.7% in VA ECMO. Including patients with clinically suspected HIT, positive ELISA, an alternative anticoagulation until discharge but missing HIPA test, the prevalence raised to 5.1% (VV ECMO 6.6%, VA ECMO 2.3%). Second, patients with confirmed HIT had higher rates of bleeding, thrombosis and technical complications compared to the ECMO control group without HIT. The nadir of the thrombocytes was less pronounced in the ECMO-control group compared to the others. However, the latter results are partly biased by the study design as group assignment was based on the 4T-score. Third, survival and neurologic outcome were similar.

The prevalence of 3.2% seems to be in contrast to the largest multicenter study evaluating HIT on VA ECMO, that reported a prevalence of 0.4% [24]. However, in comparison to patients on cardiopulmonary bypass with reported rates of 1–3% [3] and 7.3% in postcardiotomy patients requiring VA ECMO [25], the current results seem to be plausible. The applied screening algorithm in one single medical ICU in the current cohort does not suffer from different use of HIT testing and known interlaboratory variability due to the use of different functional assays [26] that might be observed in multicenter studies. Our results underline the importance of a clearly defined algorithm for HIT diagnostic.

Moreover, two other studies focussing on HIT in ECMO patients reported prevalences within a similar range of 3.1 and 8.3% [17, 18]. Taking into account the potential underestimation due to missing confirmation of HIT in the HIT suspicion group, the prevalence in our study would be higher and approach 5%.

Apparently, HIT is more common in patients undergoing ECMO therapy in comparison to critically ill patients without ECMO with rates between 0.05% and 3.1% [27, 28].

Most of the reported data included only patients requiring VA ECMO or did not differentiate the mode of ECMO [17, 18, 24]. Data on HIT in VV ECMO are scarce. In our cohort, the prevalence of HIT was trendwise higher in VV- compared to VA-ECMO. This is in contrary to the results of Pabst et al. [17] reporting a higher prevalence in VA ECMO patients. As we have seen no difference in the temporal occurrence of HIT between VV and VA ECMO patients, it is not clear yet, whether this is explained by longer duration of ECMO runs on VV ECMO in general, or by an inherent increased risk of patients supported with VV ECMO, who more often suffer from infection and sepsis. The longer ECMO duration and the difference in ventilation parameters of HIT patients in this study may also be caused by the higher proportion of VA ECMO patients in the ECMO-control group with usually shorter ECMO runs and less severe pulmonary failure and does not have to be associated with HIT. Nevertheless longer ECMO support times in this study similar to others [18] have been observed as risk factor for HIT on ECMO.

HIT results in several complications such as bleeding and thrombosis. Bleeding events were observed in 38% in the HIT-confirmed group (13 VV ECMO and 3 VA ECMO patients), less than in the study of Kimmoun et al. with 57% (only VA ECMO patients) [24]. Yet, the bleeding rate was higher in all, at least temporarily, argatroban treated groups compared to the ECMO-control group, which might be in part explained by the lower platelet counts in all HIT suspected groups. Contrary to published data, thromb-embolic events, mainly triggered by venous thrombosis, were more common in our cohort. This is in line with our previously published results with standardized and rigorous screening for venous thrombosis [6]. Overall, the lower bleeding and higher thrombotic event rates in the current study compared to the study

of Kimmoun et al. may be evoked by pre-set anticoagulation goals. In general, comparing centers with regard to bleeding and thrombosis is difficult due to varying anticoagulation strategies.

During ECMO therapy a decrease in platelet count during the first 5 days can be often observed. A secondary additional decrease is suggestive of HIT, which was confirmed in 20% of thrombocytopenias occurring later than 5 days on ECMO in our study. This emphasizes the importance of stringent testing for HIT and prompt change of anticoagulation if HIT is clinically suspected. Furthermore, in a steeper or otherwise unexplained decrease in platelet count even before the 5[th] day on ECMO, which was seen in 3 of 16 confirmed HIT patients in our study, HIT has to be considered, as most often the Heparin exposure prior to ECMO is unknown.

Early change to an alternative anticoagulation can prevent further severe complications. This was underlined by the current study as we found no excessive mortality in the HIT groups. The heparin free alternative anticoagulation on ECMO in this study was argatroban, as it is safe, easy to adjust and independent of the kidney function [21]. Changing from heparin to another anticoagulation such as argatroban may result in higher drug costs. However, overall costs of argatroban are comparable after accounting for HIT testing and complications [21, 29].

According to the HIT-4T-Score a more pronounced decrease of thrombocytes increases the probability for the confirmation of HIT [22]. Noteworthy, the decline of thrombocytes was steeper and deeper in the VA than the VV ECMO patients, but HIT was diagnosed more frequently in VV ECMO patients. The stronger decline of thrombocytes in VA might be explained by a higher turbulence in the blood circuit because of higher pressure differences in the arterial system and need for higher rotational speed of the blood pump.

After change of anticoagulation, thrombocytes increased in all groups after 3 days on argatroban. A reexposure to heparin is feared in patients with confirmed HIT when using heparin-coated ECMO circuits. We did not observe differences in changes of platelet count or complications in patients, who were supported with heparin-free systems compared to heparin-coated systems. Also, the exchange of an oxygenator to a heparin-coated system in a patient with diagnosed HIT, who was on argatroban, did not result in a drop in platelet count or complications. It is assumed, that in heparin-coated systems heparin is attached by covalent bonds to the circuit, is eluted only initially in very small amounts, rapidly covered by plasma proteins and therefore is not considered as biological active [17]. Nonetheless, while an approved systemic anticoagulant seems to be of prime importance to avoid ongoing complications, due to small numbers of patients, we cannot suggest that the use of heparin-coated circuits in patients with HIT is safe.

The overall mortality of our HIT patients was similar compared to ECMO patients without HIT and was in line with others [17, 24] despite longer requirement of ECMO support.

## Strength and limitations

This single-center retrospective study was conducted by staff with long-standing experience in the use of VV and VA ECMO and argatroban as alternative anticoagulation. Including both VV and VA ECMO patients of a single medical ICU with continuous and standardized treatment and diagnostic algorithms is one of the strength contrasting to other studies with variable protocols, diagnosis criteria or inclusion of only VA ECMO patients [20, 24].

A limitation is the retrospective design that only allows for detection of associations but not for causal relations. HIPA test was missing in the HIT-suspicion group, but an immediate response of the platelet count after the change to argatroban was seen and therefore continued

until discharge. Exposure to heparin prior to ECMO support was likely, but not known. Due to small numbers of HIT confirmation, small effects and differences between the groups may have been missed.

## Conclusions

The prevalence of confirmed HIT during ECMO for respiratory and circulatory support was 3.2%. Confirmed HIT was trendwise more frequent in VV than in VA ECMO. Complications were observed more frequently in the confirmed HIT group, but survival to discharge was similar. Further prospective multicenter studies with rigorous screening algorithms and identical diagnostic workup are warranted to gain more information and preclude over- or underdiagnosis of HIT.

## Supporting information

**S1 File. Supplementary materials.**
(PDF)

**S1 Table. Characteristics of patients with confirmed heparin-induced thrombocytopenia.**
(PDF)

**S2 Table. Localisation of thrombotic events.**
(PDF)

**S3 Table. Technical problems.**
(PDF)

**S4 Table. Bleeding Events.**
(PDF)

**S5 Table. Consumption of blood products on extracorporeal membrane oxygenation.**
(PDF)

**S1 Fig. Median D-Dimers on extracorporeal membrane oxygenation.**
(PDF)

**S2 Fig. Median fibrinogen on extracorporeal membrane oxygenation.**
(PDF)

**S3 Fig. Median activated partial prothrombin time on extracorporeal membrane oxygenation.**
(PDF)

**S4 Fig. Median plasma free hemoglobin on extracorporeal membrane oxygenation.**
(PDF)

**S5 Fig. Median antithrombin III on extracorporeal membrane oxygenation.**
(PDF)

**S6 Fig. Median C reactive protein on extracorporeal membrane oxygenation.**
(PDF)

**S7 Fig. Median leucocytes on extracorporeal membrane oxygenation.**
(PDF)

**S8 Fig. Median thrombocyte counts of veno-arterial vs. veno-venous extracorporeal membrane oxygenation of groups HIT-confirmed and HIT-suspicion.**
(PDF)

**S9 Fig. Thrombocyte counts and day of circuit exchange in individual patients of group HIT-confirmed.**
(PDF)

**S10 Fig. Thrombocyte counts and day of circuit exchange in individual patients of group HIT-suspicion.**
(PDF)

**S11 Fig. Comparison of heparin-coated ECMO-systems vs. non-heparin-coated ECMO-systems in relation to thrombocyte counts of individual patients in the group HIT-confirmed.**
(PDF)

## Acknowledgments

The authors would like to thank the nursing staff of the intensive care units and the perfusionists for their excellent patient care and ECMO control. We thank Prof. Dr. A. Greinacher and staff of the Department of Transfusion Medicine, University Medical Center Greifswald for performing the Heparin induced platelet aggregation tests.

## Author Contributions

**Conceptualization:** Matthias Lubnow, Thomas Müller.

**Data curation:** Matthias Lubnow, Johannes Berger, Roland Schneckenpointner, Florian Zeman, Alois Philipp, Maik Foltan, Susanne Heimerl.

**Formal analysis:** Matthias Lubnow, Johannes Berger, Florian Zeman, Christoph Fisser.

**Methodology:** Matthias Lubnow, Thomas Müller.

**Resources:** Alois Philipp.

**Supervision:** Thomas Müller.

**Visualization:** Johannes Berger, Florian Zeman.

**Writing – original draft:** Matthias Lubnow, Johannes Berger, Christoph Fisser.

**Writing – review & editing:** Matthias Lubnow, Johannes Berger, Roland Schneckenpointner, Florian Zeman, Dirk Lunz, Alois Philipp, Maik Foltan, Karla Lehle, Susanne Heimerl, Christina Hart, Christof Schmid, Christoph Fisser, Thomas Müller.

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
