## [Decision Letter · Decision Letter 0]

21 Jun 2022

PONE-D-21-40900Prevalence and outcomes of patients developing Heparin-induced Thrombocytopenia during extracorporeal membrane oxygenationPLOS ONE

Dear Dr. Lubnow,

Thank you for submitting your manuscript to PLOS ONE. After careful consideration, we feel that it has merit but does not fully meet PLOS ONE’s publication criteria as it currently stands. Therefore, we invite you to submit a revised version of the manuscript that addresses the points raised during the review process.

We look forward to receiving your revised manuscript.

Kind regards,

Cécile Oury

Academic Editor

PLOS ONE

Journal Requirements:

I have read the journal's policy and the authors of this manuscript have the following competing interests: ML recieved lecture honoraries from Fresenius Medical Care.

Additional Editor Comments:

Some clarifications and minor revision are needed as per reviewer's comments.

Reviewers' comments:

Reviewer's Responses to Questions

**Comments to the Author**

1. Is the manuscript technically sound, and do the data support the conclusions?

Reviewer #1: Yes

2. Has the statistical analysis been performed appropriately and rigorously? 

Reviewer #1: Yes

3. Have the authors made all data underlying the findings in their manuscript fully available?

Reviewer #1: Yes

4. Is the manuscript presented in an intelligible fashion and written in standard English?

Reviewer #1: Yes

5. Review Comments to the Author

Reviewer #1: In this manuscript, Lubnow et al. describe the single centre results for prevalence and outcome of HIT in patients on ECMO.

The study is well designed, well performed and well described.

Remarks:

-The HIT-suspicion group consists of patients with high 4T-score, positive ELISA but missing HIPA test. The authors should explain why the HIPA test was not performed in these samples? If the HIPA test would have been performed, all samples could be attributed to either the HIT-confirmed or the HIT-excluded group. In this regard, one would expect that results of all outcome parameters for the HIT-suspicion group would fall between the HIT-confirmed and the HIT-excluded group. However, the HIT-suspicion group seems to differ from both other groups with often the highest or lowest results for e.g. gender, BMI, days on ECMO, days from ECMO start tot HIT suspicion (table 2), starting value for D-dimers (fig S1), fibrinogen (fig S2), CRP (fig S6) and leukocytes (fig S7). How can this observation be explained?

-Some interpretations of comparison between groups (e.g. p10: 'PLT nadir was lower in groups with suspicion of HIT compared to control' or p15: 'patients with confirmed HIT had higher rates of bleeding and thrombosis compared to the control group') result directly from the study design, as groups were initially defined based on the 4T-score. This should at least be discussed by the authors.

-Table 2: the indices '1' (HIT vs control) and '4' (confirmed HIT vs HIT supicion) are lacking in the table.

-M&M and Table 2: 185 patients from the control group were excluded from the analysis according to Table2, and 251 were included. The sum of both (436) does not agree with the number of 426 in Table1.

-p19: the authors seem to insinuate that the rise in PLT upon switching to argatroban is an argument to consider the HIT-suspicion group as 'real HIT', even if the HIPA-test was not performed. However, an exact same rise in PLT was observed in the HIT-excluded group.

-Fig3 is difficult to understand. How should the Y-axis be interpreted?

6. PLOS authors have the option to publish the peer review history of their article (what does this mean?). If published, this will include your full peer review and any attached files.

Reviewer #1: No

---

## [Author Response · Author response to Decision Letter 0]

11 Jul 2022

Response to the editor and the reviewer

Please use the uploaded version of this response letter.

Manuscript ‘Prevalence and outcomes of patients developing Heparin-induced Thrombocytopenia during extracorporeal membrane oxygenation’

We very much appreciate the thoughtful and constructive comments. Please find our specific responses below. We modified the manuscript according to your suggestions. 

Journal Requirements:

C1: Please ensure that your manuscript meets PLOS ONE's style requirements, including those for file naming. The PLOS ONE style templates can be found at 

R1: The main manuscript and all supplementary information have been revised according to the PLOS ONE requirements.

C2: Thank you for stating the following in the Competing Interests section: 

I have read the journal's policy and the authors of this manuscript have the following competing interests: ML received lecture honoraria from Fresenius Medical Care.

C3: We note that you have indicated that data from this study are available upon request. PLOS only allows data to be available upon request if there are legal or ethical restrictions on sharing data publicly. For more information on unacceptable data access restrictions, please see http://journals.plos.org/plosone/s/data-availability#loc-unacceptable-data-access-restrictions. 

R2+3: The cover letter has been revised according to the PLOS ONE requirements.

C4: Please review your reference list to ensure that it is complete and correct. If you have cited papers that have been retracted, please include the rationale for doing so in the manuscript text, or remove these references and replace them with relevant current references. Any changes to the reference list should be mentioned in the rebuttal letter that accompanies your revised manuscript. If you need to cite a retracted article, indicate the article’s retracted status in the References list and also include a citation and full reference for the retraction notice.

R4: We reviewed our reference list. It is complete and correct. No cited papers have been retracted.

Additional Editor Comments:

Some clarifications and minor revision are needed as per reviewer's comments.

Reviewers' comments:

Reviewer's Responses to Questions

Comments to the Author

1. Is the manuscript technically sound, and do the data support the conclusions?

Reviewer #1: Yes

2. Has the statistical analysis been performed appropriately and rigorously? 

Reviewer #1: Yes

3. Have the authors made all data underlying the findings in their manuscript fully available?

Reviewer #1: Yes

4. Is the manuscript presented in an intelligible fashion and written in standard English?

Reviewer #1: Yes

5. Review Comments to the Author

Reviewer #1: In this manuscript, Lubnow et al. describe the single centre results for prevalence and outcome of HIT in patients on ECMO.

The study is well designed, well performed and well described.

Remarks:

C5: -The HIT-suspicion group consists of patients with high 4T-score, positive ELISA but missing HIPA test. The authors should explain why the HIPA test was not performed in these samples? If the HIPA test would have been performed, all samples could be attributed to either the HIT-confirmed or the HIT-excluded group. In this regard, one would expect that results of all outcome parameters for the HIT-suspicion group would fall between the HIT-confirmed and the HIT-excluded group. However, the HIT-suspicion group seems to differ from both other groups with often the highest or lowest results for e.g. gender, BMI, days on ECMO, days from ECMO start tot HIT suspicion (table 2), starting value for D-dimers (fig S1), fibrinogen (fig S2), CRP (fig S6) and leukocytes (fig S7). How can this observation be explained?

R5: The group HIT-suspicion consists of patients with positive ELISA but without a HIPA test which were switched to argatroban because of suspected HIT. Clinically they were treated like HIT patients and during the initial design of the study we wanted to combine the groups HIT-confirmed and HIT-suspicion, which of course scientifically is not correct and therefore we made the 2 groups. The reason for the absence of the HIPA test was usually that it was forgotten to be sent or that it was temporarily unavailable, therefore no specific preselection of this patient group has occurred. It just happened by chance. Thus there is no good explanation for the differences. Even though the patients in group HIT-suspicion had a numerically higher BMI, were intubated later, had a more aggressive ventilation, more inflammation and activation of the coagulation which might result in longer ECMO runs, numerically longer ICU and hospital stays, but numbers are low, differences were mostly not statistically significant and if, alternating between the different ECMO groups. Thus we thought it is of minor importance and haven’t discussed it.

We added to Materials and methods / section Screening algorithm for heparin-induced thrombocytopenia:

The reason for the absence of the HIPA test in the HIT-suspicion group was usually that it was forgotten to be sent or that it was temporarily unavailable, therefore no specific preselection of this patient group has occurred.

C6: -Some interpretations of comparison between groups (e.g. p10: 'PLT nadir was lower in groups with suspicion of HIT compared to control' or p15: 'patients with confirmed HIT had higher rates of bleeding and thrombosis compared to the control group') result directly from the study design, as groups were initially defined based on the 4T-score. This should at least be discussed by the authors.

R6: We agree with the reviewer that this has to be mentioned and complemented the Results and discussion section accordingly

P10: At admission, platelets were not different between groups, but the nadir was significantly lower in those groups with initial suspicion of HIT compared to the ECMO control group, which results also from the study design as groups were initially defined based on the 4T-score.

P15: Second, patients with confirmed HIT had higher rates of bleeding, thrombosis and technical complications compared to the ECMO control group without HIT. The nadir of the thrombocytes was less pronounced in the ECMO-control group compared to the others. However, the latter results are partly biased by the study design as group assignment was based on the 4T-score.

C7: -Table 2: the indices '1' (HIT vs control) and '4' (confirmed HIT vs HIT supicion) are lacking in the table.

R7: That is true, we wanted to use the same indices in Table 1 and 2 but missing significant differences for indices 1 and 4 may be confusing, therefore we changed the indices as follows:

Post-hoc pairwise comparisons with significance (p<0.05): 1 HIT suspicion vs. ECMO control; 2 excluded HIT vs. ECMO control; 3 HIT suspicion vs. excluded HIT.

C8: -M&M and Table 2: 185 patients from the control group were excluded from the analysis according to Table2, and 251 were included. The sum of both (436) does not agree with the number of 426 in Table1.

R8: We agree with the reviewer and are very sorry. This is a calculation error. 426 (ECMO control group patients in total) – 251 (ECMO control group patients with thrombotic complications screening) = 175 patients which had been excluded.

We changed the caption of Table 2 accordingly:

175 patients from group ECMO control were excluded in this analysis, because standardized screening for thrombotic complications without any HIT suspicion was only established in 2014.

C9: -p19: the authors seem to insinuate that the rise in PLT upon switching to argatroban is an argument to consider the HIT-suspicion group as 'real HIT', even if the HIPA-test was not performed. However, an exact same rise in PLT was observed in the HIT-excluded group.

R9: We somewhat disagree with the reviewer. The Hit-suspicion group (yellow line) starts on day x+2 from the lowest median platelet count and rises to day x+7 to the highest median thrombocyte count (as high as the ECMO-control group, nearly normal) whereas the median platelet count of the HIT excluded group (green line) only slightly rises in parallel with the HIT-confirmed group. That rise in platelet count was the reason to remain on argatroban even with missing HIPA test in group HIT-suspicion. According to these reasons no changes were made in the manuscript.

C10: -Fig3 is difficult to understand. How should the Y-axis be interpreted?

R10: This Kaplan Meier plot shows the temporal relationship of the HIT-occurrence in VV and VA ECMO patients. It shows that the HIT diagnosis took place early in both groups. Though the longer ECMO duration of the VV ECMO patients is probably not the reason for a higher HIT prevalence. 

We agree with the reviewer and complemented the caption of Fig 3 for a better understanding:

Fig 3. Confirmed HIT probability during ECMO therapy according to VV and VA ECMO

Heading: Time to HIT in VV and VA ECMO patients of group HIT-confirmed

Kaplan-Meier plot of HIT probability during ECMO therapy according to VV ECMO and VA ECMO of group HIT-comfirmed. X-axis: Time to HIT = Time between ECMO initiation and switch to alternative anticoagulation (argatroban) in days. Y-axis: Probability of confirmed HIT diagnosis 0-100%.

---

## [Editor Report · Decision Letter 1]

22 Jul 2022

Prevalence and outcomes of patients developing Heparin-induced Thrombocytopenia during extracorporeal membrane oxygenation

PONE-D-21-40900R1

Dear Dr. Lubnow,

We’re pleased to inform you that your manuscript has been judged scientifically suitable for publication and will be formally accepted for publication once it meets all outstanding technical requirements.

Kind regards,

Cécile Oury

Academic Editor

PLOS ONE

Additional Editor Comments (optional):

The authors have adequately answered all reviewer's comments.
---

## [Editor Report · Acceptance letter]

29 Jul 2022

PONE-D-21-40900R1 

Prevalence and outcomes of patients developing heparin-induced thrombocytopenia during extracorporeal membrane oxygenation 

Dear Dr. Lubnow:

I'm pleased to inform you that your manuscript has been deemed suitable for publication in PLOS ONE. Congratulations! Your manuscript is now with our production department. 

Kind regards, 

on behalf of

Dr. Cécile Oury 

Academic Editor

PLOS ONE